

# Multidimensional evaluation of the TRMM 3B43V7 satellite-based precipitation product in mainland China from 1998–2016

Ziteng Zhou[1,2], Bin Guo[1,2], Youzhe Su[1,2], Zhongsheng Chen[3] and Juan Wang[4]

[1] Key Laboratory of Geomatics and Digital Technology of Shandong Province, Shandong University of Science and Technology, Qingdao, Shandong, China
[2] College of Geomatics, Shandong University of Science and Technology, Qingdao, Shandong, China
[3] School of Land and Resources, China West Normal University, Nanchong, Sichuan, China
[4] College of Science and Information, Qingdao Agricultural University, Qingdao, Shandong, China

## ABSTRACT

This study evaluates the applicability of the Tropical Rain Measurement Mission (TRMM) 3B43V7 product for use throughout mainland China. Four statistical metrics were used based on the observations made by rain gauges; these metrics were the correlation coefficient ($R$), the relative bias ($RB$), the root mean square error ($RMSE$) and Nash-Sutcliffe efficiency ($NSE$), and they were chosen to evaluate the performance of the 3B43V7 product at temporal and spatial scales. The results revealed that 3B43V7 performed satisfactorily on all timescales ($R > 0.9$ and $NSE > 0.86$); however, it overestimated the results when compared with the rain gauge observations in certain circumstances ($RB = 9.7\%$). Monthly estimates from 3B43V7 were in agreement with rain gauge observations. 3B43V7 can effectively capture the seasonal patterns of precipitation characteristics over mainland China. However, 3B43V7 tends to register a greater overestimation of precipitation in the winter ($RB = 14\%$) than in other seasons while showing greater consistency with the observations made by rain gauges during dry periods. The 3B43V7 product performs well in the eastern part of mainland China, while its performance is poor in the western part of mainland China. In terms of altitude, 3B43V7 performs satisfactorily in areas with moderate to low altitudes (when altitude $< 3,500$ m, $R > 0.9$, $NSE > 0.8$ and $RB < 10.2\%$) but $RB$ values increase with altitude. Overall, 3B43V7 had a favorable performance throughout mainland China.

# INTRODUCTION

Precipitation is vitally important to the global water cycle and energy balance. Precise precipitation data at high spatial and temporal resolutions are essential for hydrological research, water resources management, agricultural production, drought and flood early warnings, and for monitoring purposes (*Chen & Li, 2016*; *Chen et al., 2013b*; *Habib et al., 2012*; *Seyyedi et al., 2015*; *Tang et al., 2015*; *Zhao et al., 2018*). Precipitation data are traditionally acquired from in situ rain gauge stations and weather radars. Rain gauges can

Corresponding author
Bin Guo, guobin07@mails.ucas.ac.cn

directly obtain precipitation information at specific sites, which is regarded as the more reliable means for observing precipitation (*Xie & Arkin, 1996*). However, the networks of rain gauges are often sparse and unevenly distributed because of the limited access to certain areas by meteorological stations (*Maggioni, Meyers & Robinson, 2016*; *Zhu et al., 2017*). Weather radars also suffer from limited spatial coverage, particularly in oceanic, remote, or undeveloped regions. These shortcomings of rain gauge stations and weather radars limit their applications in remote areas because the spatial precipitation data are not accurate in these regions (*Ebert, Janowiak & Kidd, 2007*; *Kidd et al., 2012*; *Ma et al., 2016*; *Porcù, Milani & Petracca, 2014*). Therefore, many researchers are looking for alternatives to overcome these limitations (*Li et al., 2008*; *Liu, 2015*).

Compared to rain gauge observations, satellite sensors can access difficult-to-reach regions to generate a quasi-global coverage of precipitation estimates with adequate temporal resolution and fine spatial resolution (*Javanmard et al., 2010*; *Jiang et al., 2012*; *Mantas et al., 2015*; *Shrivastava et al., 2014*). Currently, a number of satellite-based precipitation products are available to the public, which mainly includes the Precipitation Estimation from Remotely Sensed Information using Artificial Neural Network (PERSIANN) (*Moazami et al., 2013*), the Climate Precipitation Center morphing technique (COMRPH) (*Habib et al., 2012*), the Tropical Rainfall Measure Mission (TRMM) Multi-satellite Precipitation Analysis (TMPA) (*Liu, 2015*), and the Integrated Multi-Satellite Retrievals for Global Precipitation Measurement (GPM) mission (IMERG) (*Villarini et al., 2008*). TRMM was developed jointly by the National Aeronautics and Space Administration (NASA) and the Japanese Aerospace Exploration Agency (JAXA) in 1997. TRMM was retired on 8 April 2015 after more than 17 years of productive data collection. These products provided important precipitation information for scientific studies (*Huffman et al., 2007*; *Kummerow et al., 1998*; *Kummerow et al., 2001*; *Mehta & Yang, 2008*; *Yang & Smith, 2008*). The accuracy of satellite-based precipitation estimates has been a recent focus of attention due to the temporal and spatial variabilities of precipitation (*Chen & Li, 2016*; *Chen et al., 2013a*; *Guo et al., 2016*; *Li, Zhang & Xu, 2012*; *Mei et al., 2014*; *Shen et al., 2014*; *Zhao & Yatagai, 2014*). Satellite precipitation products have registered deviations and random errors caused by various factors such as uncertainty in the sampling frequency and precipitation inversion algorithms (*Nair, Srinivasan & Nemani, 2010*). Therefore, satellite precipitation products must be verified by conventional rain gauge data before they can be applied.

Previous studies evaluated the performance of TMPA products using rain-gauge networks at different scales around the world (*Darand, Amanollahi & Zandkarimi, 2017*; *Guo et al., 2016*; *Hashemi et al., 2017*; *Huffman et al., 1997*; *Tang, Long & Hong, 2016a*; *Tang et al., 2016b*; *Zhao et al., 2015*). Some previous studies related to the evaluation and applications of TRMM 3B43 product are summarized in Table 1.

In summary, many such studies focused on evaluating the TRMM 3B43V7 product in the basin or on regional scales in China. However, the performance of TRMM 3B43V7 product varied greatly due to diverse altitudes and geographic locations (*Yang et al., 2017*). China is a country with frequent drought and flood disasters. The performance of TRMM 3B43 is very important for drought prediction and assessment. Therefore, a more comprehensive

Zhou et al. (2020), *PeerJ*, DOI 10.7717/peerj.8615

**Table 1  Selected studies related to the evaluation and application of TRMM 3B43.**

| Reference | Study area | Number of stations | Study period | Main findings |
|---|---|---|---|---|
| *Tan et al. (2018)* | Singapore | 22 | 1998–2014 | TRMM 3B43 outperforms NCEP-CFSR in drought monitoring over Singapore. |
| *Karaseva, Prakash & Gairola (2012)* | Kyrgyzstan | 35 | 1998–2007 | TRMM-3B43 product has statistically significant correlation with rain gauge data over the most parts of the country. |
| *Zhu et al. (2017)* | Southwest monsoonregion of China | 46 | 1998–2011 | TRMM 3B43 data and observational data have a strong correlation, but the TRMM 3B43 precipitation data are consistently lower than that obtained from the weather stations. |
| *Tan et al. (2017)* | Kelantan River Basin, Malaysia | 42 | 1998–2014 | TRMM 3B43 performs well in the monthly precipitation estimation, but performs moderately in the seasonal scale. |
| *Darand, Amanollahi & Zandkarimi (2017)* | Iran | 157 | 2003–2009 | The spatial and temporal variations of precipitation are well captured by the multi-satellite product TMPA. |
| *Wang et al. (2017)* | Qinling-Daba Mountains | 27 | 2000–2014 | TRMM 3B43 agrees with ground observations in autumn. |
| *Jin, Zhang & Huang (2015)* | Yangtze River Basin | 224 | 2003–2010 | The performance of TRMM 3B43 during the wet period is better than in the dry period. |
| *Tao et al. (2016)* | Jiangsu Province, China | 65 | 1998–2014 | TRMM 3B43 precipitation data could be used for reliable short-term drought monitoring in Jiangsu Province. |
| *Chen & Li (2016)* | Mainland China | 750 | 2014–2015 | IMERG performs a little better than TRMM 3B43 at seasonal and monthly scales. |
evaluation of TRMM 3B43V7 product is urgently needed throughout mainland China, which will provide a strong theoretical support for the application of TRMM 3B43 in drought monitoring and warning in China.

The primary objective of this study was to systematically evaluate the accuracy of the TRMM 3B43V7 product over mainland China using observations from 620 meteorological stations from January 1998 to December 2016. The present study emphasizes multiple spatial and temporal scales as well as altitude as a factor. The results of the study can provide theoretical support for TRMM 3B43V7 to monitor and evaluate drought. The use of TRMM 3B43V7 can improve the ability to monitor drought conditions in complex terrains, especially in western regions of China where rain gauge observations are lacking. The study is organized into the study area with data and statistic metrics ('Materials & Methods'), results and discussion ('Results and Discussion') and our conclusions ('Conclusions').

## MATERIALS & METHODS

### Study area

The study was conducted over mainland China, located within $73°41'–135°02'E$ and $18°10'–53°33'N$. The geography of mainland China is variable and the topography has obvious regional differences. Its terrain is gradually reduced from the northwest to the southeast (Fig. 1A), and its climate is highly complex because of the confluence of different climatological factors. These factors cause the spatiotemporal variability of annual precipitation over mainland China. The annual precipitation gradually decreases from the southeast to the northwest (Fig. 1B). In order to evaluate the performance of TRMM 3B43V7 in different regions of China, mainland China is further divided into nine basins (Fig. 1A) according to the principles provided by the Resource and Environment Data Center of the Chinese Academy of Sciences for the Division of Water Resources in China (http://www.resdc.cn). The basin name, the number of meteorological stations and major climate type in each basin are listed in Table 2. The climate conditions in some regions of mainland China are harsh because of the various physical conditions, especially in the western inland area. There are few meteorological stations in the desert and alpine regions of western China that have higher altitudes with a greater distribution in the lower altitude areas of eastern China. In contrast to the varied placement of meteorological stations, the uninterrupted data from the 3B43V7 product has an unparalleled advantage.

### Dataset
#### TRMM 3B43V7
TRMM was launched in late 1997 by the National Aeronautics and Space Administration (NASA) and the National Space Development Agency of Japan (NASDA) with the main objective to monitor tropical and subtropical precipitation. The TRMM Multi-satellite Precipitation Analysis (TMPA) algorithm has undergone three major upgrades, which are attributed to new sensors and algorithmic changes, during the last 10 years. The latest version of TMPA algorithm is Version 7, which mainly includes 3B42V7, 3B42RTV7 and 3B43V7 satellite precipitation estimation products. Among them, 3B43V7 was used in

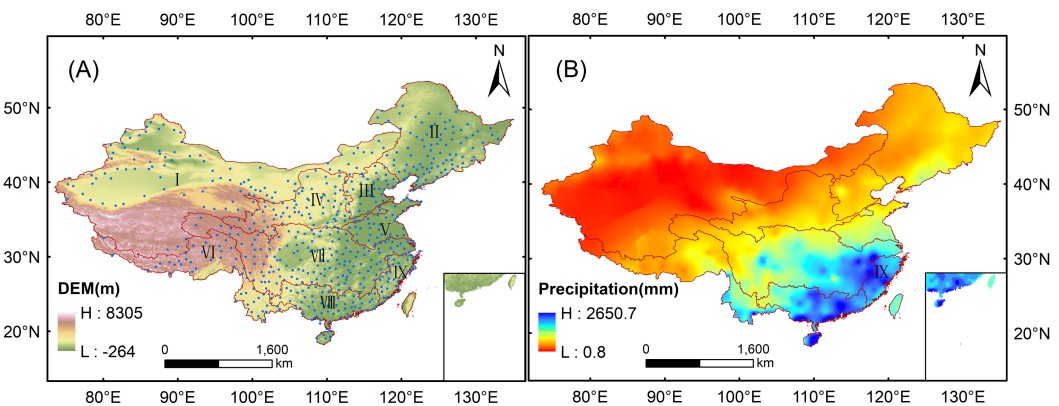

**Figure 1** Study area of the evaluation of the 3B43V7 product: (A) terrain altitude and nine basins; (B) average annual precipitation (1998–2016).

**Table 2** Basin name, the number of meteorological stations and major climate type in each basin.

| River basin name | Number of stations | Major climate type |
|---|---|---|
| (I) Inland River Basin | 97 | Temperate continental climate |
| (II) Songliao River Basin | 92 | Temperate continental monsoon climate |
| (III) Haihe River Basin | 32 | Temperate East Asian monsoon climate |
| (IV) Yellow River Basin | 70 | Temperate monsoon climate |
| (V) Huaihe River Basin | 37 | Temperate and subtropical monsoon climate |
| (VI) Southwest River Basin | 44 | Subtropical monsoon climate |
| (VII) Yangtze River Basin | 148 | Subtropical monsoon climate |
| (VIII) Pearl River Basin | 66 | Subtropical monsoon climate |
| (IX) Southeast River Basin | 34 | Subtropical monsoon climate |

this present study. The 3B43V7 product provides a monthly estimate of precipitation with a spatial resolution of 0.25°, covering the global region between 50°S and 50°N (most of mainland China). This dataset is generated by combining different products from the precipitation radar (PR) instrument and the TRMM microwave imager (TMI) instrument with additional data from other sources to include geosynchronous infrared, SSM/I microwave, and rain gauges. The TRMM 3B43V7 is mainly computed in four stages: (1) the microwave precipitation estimates are inter-calibrated and combined, (2) IR precipitation estimates are created using the calibrated microwave precipitation, (3) the microwave and IR estimates are combined, and (4) rain gauge data are integrated. The TRMM 3B42 lacks the fourth step and has a few simplifications. More details about TRMM rainfall products can be found at *Huffman et al. (2010)*; *Huffman et al. (2007)*. Routine production of TRMM PR precipitation estimates ended on October 7, 2014 due to the continuous descent and final decommissioning of the TRMM satellite. The climatological calibrations and adjustments have been adopted for TMPA products since
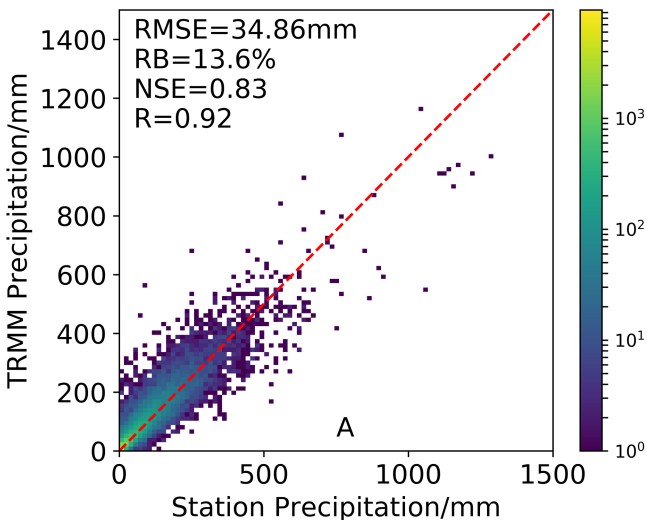

**Figure 2** Accuracy evaluation of TRMM 3B43V7 grids without gauges of GPCC.

that time and after more than 17 years of data collection, the instruments on TRMM were turned off on 8 April 2015. However, the dataset was produced until 2018.

This study used the monthly data from 3B43V7 from January 1998 to December 2016. The data were obtained from the Precipitation Measurement Mission website (https://pmm.nasa.gov/data-access/downloads/trmm). A total of 620 grid boxes were selected within mainland China. These grid boxes were selected based on the requirement that they contain at least one meteorological station within the area of the grid box to determine the validation against the observed precipitation date from rain gauges. Additionally, the annual and seasonal 3B43V7 precipitation data were computed by accumulating monthly datasets.

In addition, the readers should bear in mind that the rain gauge data used in TRMM 3B43V7 are not be completely independent from those used in this study, which could be make the validation results unstable in mainland China. To prove the credibility of our results, we first exclude all grid cells with rain gauge data correction of GPCC for a "worst case" comparison. In this case, if TRMM 3B43V7 still has consistent performance, then we can conclude that "the results are credible". Figure 2 shows that the TRMM 3B43V7 as a whole still has perfect performance when the data is completely independent ($R = 0.92$, $RMSE = 34.86$, $RB = 13.6\%$ and $NSE = 0.83$). Therefore, we think that our results are credible.

## Rain gauge data

To evaluate the performance of the TRMM 3B43V7 precipitation product, comparisons were made against the observed precipitation data recorded by the rain gauges. The monthly precipitation observations of 756 meteorological stations were provided by the National Meteorological Information Center of the China Meteorological Administration and were strictly controlled for quality (https://data.cma.cn). In terms of quality control, three types of quality control methods (i.e., climate limit value check, station extreme value check,

and internal consistency check) were used for quality inspection and control of the data (*CMA, 2007*). The data in this study were collected from January 1998 to December 2016. Due to chronic lack of measurement data at some stations, temporal consistency control using data length was conducted and 620 meteorological stations with relatively complete measurements were selected. The rain gauge used by the meteorological station is mainly a tilting rain gauge. Tilting rain gauge is an important rainfall observation instrument that can observe rainfall data digitization in real time. It has the advantages of accurate time, automatic data recording, and convenient data collection and processing, and it can record snowfall (*CMA, 2007*; *WMO, 2008*; *WMO, 2015*). The majority of the meteorological stations were located in the eastern and central parts of mainland China (Fig. 1), with only a sparse distribution over the western part, especially the western part of the Tibetan Plateau. A more direct comparison was made in this study, in which the stations within a grid box were extracted and matched to the box; for any grid box containing more than one station, the average precipitation of those stations was compared with the satellite estimate in that grid box. The annual and seasonal precipitation values were accumulated from monthly observations to validate the TRMM 3B43V7 product.

## Methodology

Four statistics metrics were used to evaluate the performance of the 3B43V7 product versus the observed precipitation from the rain gauges including the correlation coefficient ($R$), root mean square error ($RMSE$), relative bias (RB) and Nash-Sutcliffe efficiency ($NSE$). These metrics were calculated as follows:

$$R = \frac{\sum_{i=1}^{n}(G_i - \bar{G})(S_i - \bar{S})}{\sqrt{\sum_{i=1}^{n}(G_i - \bar{G})^2}\sqrt{\sum_{i=1}^{n}(S_i - \bar{S})^2}} \tag{1}$$

$$RMSE = \sqrt{\frac{1}{n}\sum_{i=1}^{n}(S_i - G_i)^2} \tag{2}$$

$$RB = \frac{\sum_{i=1}^{n}(S_i - G_i)}{\sum_{i=1}^{n}G_i} \times 100\% \tag{3}$$

$$NSE = 1 - \frac{\sum_{i=1}^{n}(G_i - S_i)^2}{\sum_{i=1}^{n}(G_i - \bar{G})^2} \tag{4}$$

where $S_i$ and $G_i$ are the values of the satellite precipitation data and rain gauge observations for the $i$th rain station, respectively; $\bar{S}$ and $\bar{G}$ are the mean values of the satellite precipitation data and rain gauge observations, respectively; and $n$ is the total number of rain gauges.

$R$ describes the linear agreement between TRMM 3B43V7 precipitation estimates and rain gauge observations. *RMSE* measures the absolute error between the TRMM data and the true value. *RB* is used to denote the degree of overall overestimation or underestimation between the satellite-based precipitation estimates and reference ground-based data. *NSE* = 1 means that the product agrees exactly with the station observations, while *NSE* = 0 indicates that the mean square error of the product is as large as just using the mean observed value as the predictor.

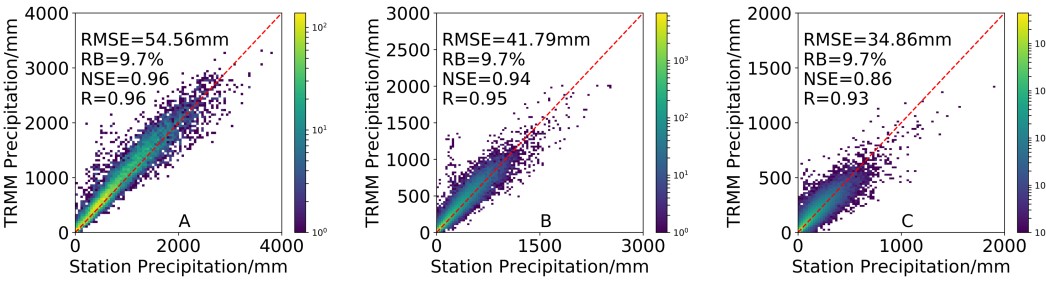

**Figure 3** Density-colored scatterplots of (A) annual, (B) seasonal and (C) monthly precipitation from 3B43V7 product versus rain gauge observations during 1998–2016. All results have passed the significance test.

## RESULTS AND DISCUSSION

### Temporal evaluation of precipitation product

A comparison of the density-colored scatterplots from 3B43V7 and the observations made by rain gauges over different timescales from January 1998 to December 2016 is shown in Fig. 3. Statistic metrics for each timescale are also indicated in the plots. These scatterplots show that the 3B43V7 product performs well, as indicated by the statistics metrics and the points located closer to the diagonal 1:1 line. The annual estimate from 3B43V7 has the best agreement with gauge observations in terms of precipitation amounts ($R = 0.96$ and $NSE = 0.96$). This is largely due, in part, to errors in the monthly estimates that cancel out when aggregated to the annual values (*Villarini & Krajewski, 2007*). Overall, the performance of the 3B43V7 product improves substantially with temporal aggregation (*Chen et al., 2013b; Meng et al., 2014*). The results obtained are consistent with those of *Mantas et al. (2015)* in the Peruvian Andes and *Darand, Amanollahi & Zandkarimi (2017)* in Iran. The values from *RMSE* at annual, seasonal, and monthly timescales were 54.56 mm, 41.79 mm, and 34.86 mm, respectively. In addition, the 3B43V7 product overestimates precipitation over the annual, seasonal, and monthly timescales with an *RB* of 9.7%.

The density-colored scatterplots from the results of 3B43V7 are compared against the observations taken from the rain gauges in the four seasons in Fig. 4. 3B43V7 can effectively capture the seasonal patterns of precipitation characteristics over mainland China. 3B43V7 has high *R* and *NSE* values in all seasons. The maximum *RMSE* and minimum *NSE* appear in summer ($RMSE = 33.44$ mm and $NSE = 0.82$). In terms of *RB*, 3B43V7 overestimates precipitation in all seasons, especially in winter ($RB = 14.0\%$). This may be the result of snowfall, which is a factor that seriously affects the precipitation estimates made by satellites in North China (*Chen & Li, 2016*). *Yong et al. (2016)* also speculated that overestimation of winter precipitation can be attributed to the inability of TRMM-based constellation satellites to measure snowfall, or the low-level cloudiness and warm rain processes in winter that may not have a strong signature of ice particles. The seasonal differences in precision for satellite-based products may be related to strong precipitation events that can be easily detected by PMW sensors during the rainy season and the accuracy of retrievals that are hampered by ice and snow cover in the winter (*Ma et al., 2015*). In addition, passive

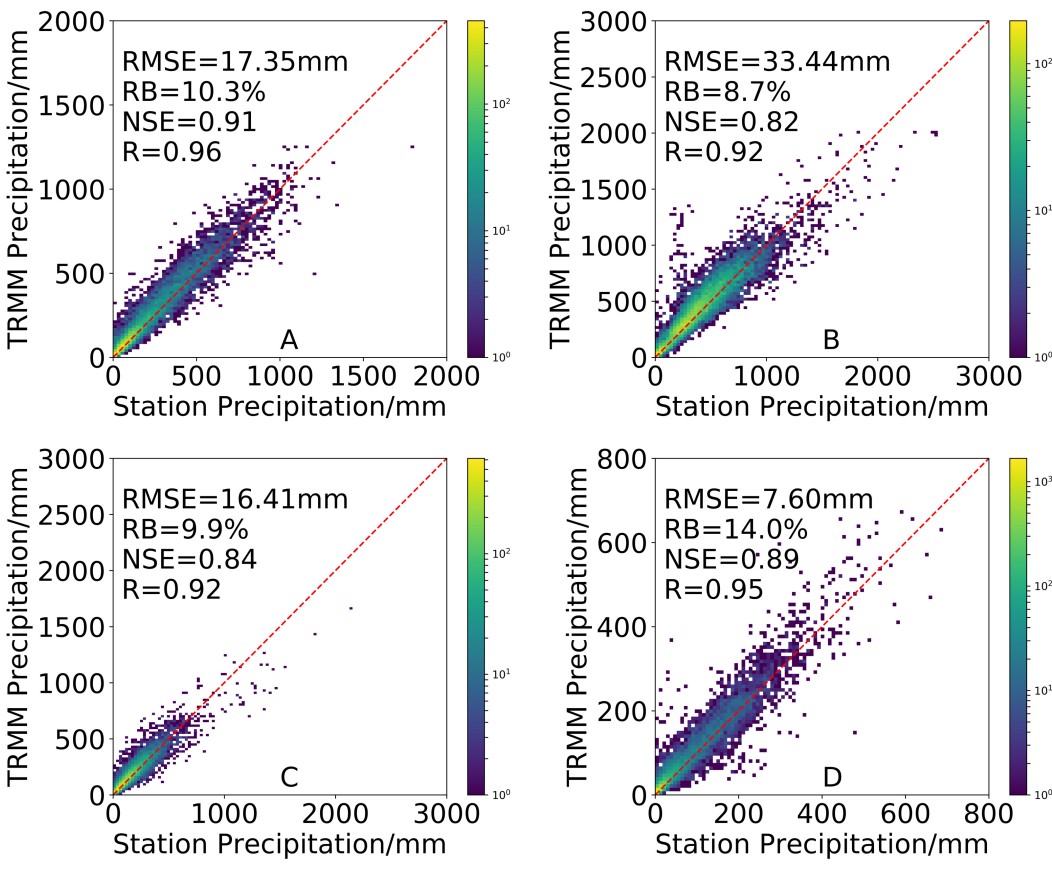

**Figure 4** Density-colored scatterplots of seasonal precipitation from the 3B43V7 product versus rain gauge observations for (A) spring, (B) summer, (C) autumn and (D) winter (All results have passed the significance test).

microwave sensors using high-frequency channels might detect an increased amount of scattering associated with the frozen surface of the land and ice particles in winter (*Chen & Li, 2016*; *Huffman & Bolvin, 2013*).

Figure 5 shows the accuracy of 3B43V7 from January to December. The results revealed that 3B43V7 had a reasonable monthly performance according to the *RB* (from 8.0% in August to 15.7% in January), *R* (from 0.87 in November to 0.95 in January), *RMSE* (from 3.35 mm in December to 16.67 mm in July), and *NSE* (from 0.74 in November to 0.95 in March). *R* values for 3B43V7 in all months were more than 0.87, while *R* is mostly below 0.9 in July–September and November. Similar to *R*, 3B43V7 also has lower *NSE* values in July-September (*NSE* < 0.8). 3B43V7 overestimates the precipitation by more than 8.0% on all monthly timescales. A higher *RB* (above 10%) occurs from November to April. In terms of *RMSE*, the values are smaller in these months. These results clearly demonstrate that there is greater consistency between 3B43V7 and the precipitation observed by a conventional rain gauge during the dry periods. This is similar to the results of *Zhao & Yatagai (2014)*. They found that the *RMSE* of the monthly precipitation recorded by both the satellite observations and the rain gauges was less for the dry season.

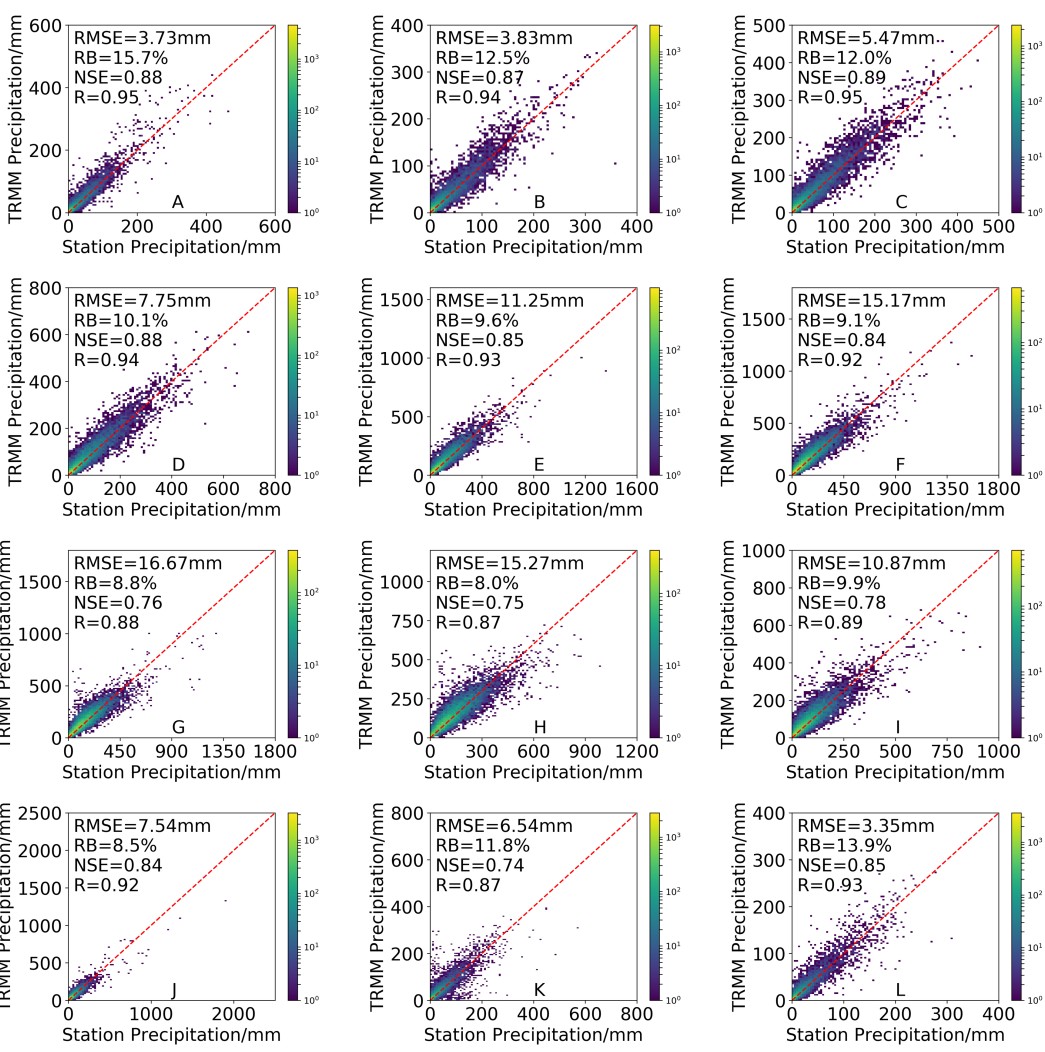

**Figure 5** Density-colored scatterplots of monthly precipitation from the 3B43V7 product versus rain gauge observations from January to December (A–L) during 1998–2016 (All results have passed the significance test).

## Spatial distributions of statistical metrics

Figure 6 shows the spatial distributions of *R* and *RMSE* between 3B43V7 and the rain gauge observations at the annual, seasonal, and monthly timescales. The performance of 3B43V7 presents obvious and significant spatial discrepancies. The spatial distributions of *R* for annual, seasonal, and monthly timescales show similar patterns over mainland China, decreasing gradually from the southeast to the northwest. High *R* values are generally found over most of eastern China, especially over southeast China. However, there are weak correlations in semi-arid and arid regions, particularly in the Tibetan Plateau and Tarim Basin, which is consistent with the results of *Chen & Li (2016)*. This may be attributed to the fact that complex topographic conditions and climates pose a great challenge for estimating accurate precipitation amounts from TRMM or that the number of gauges used
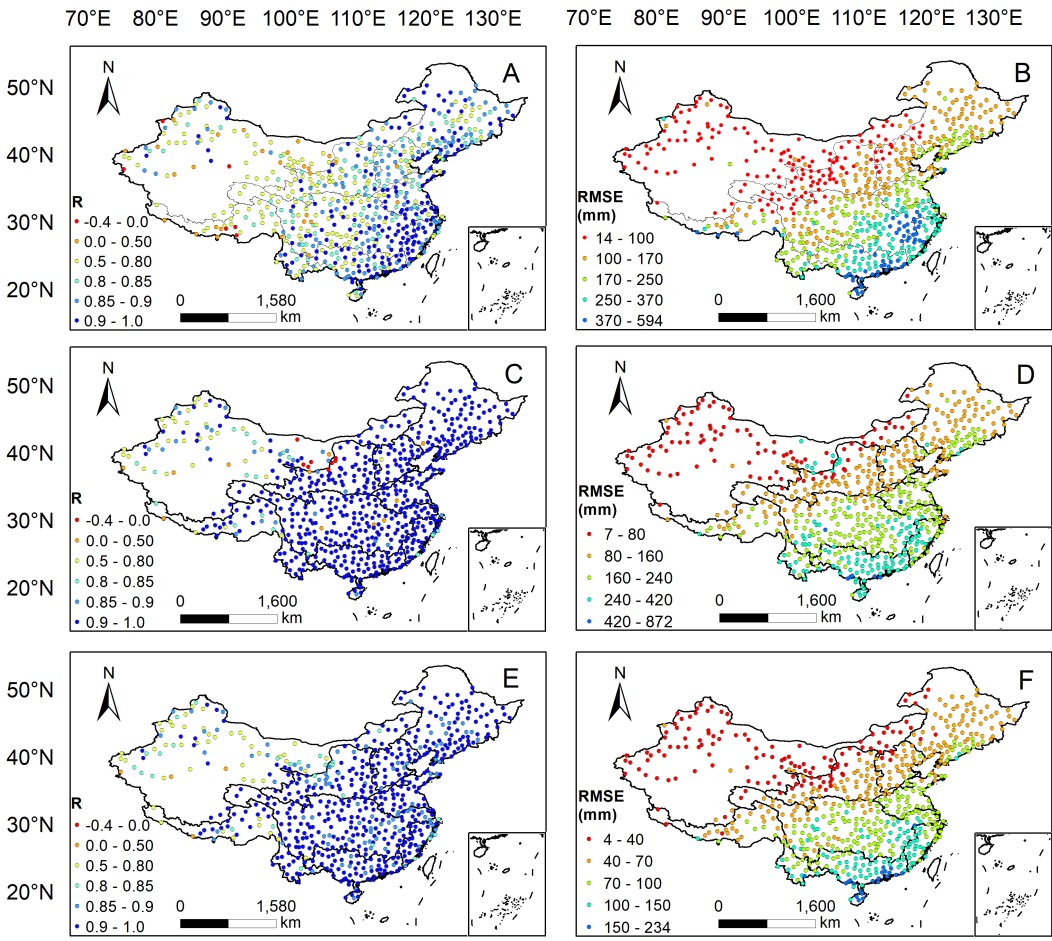

**Figure 6** Spatial distributions of *R* and *RMSE* between the 3B43V7 product and rain gauge observations at the annual (A–B), seasonal (C–D) and monthly (E–F) scales.

in the GPCC product was limited in these regions (*Huang et al., 2016*). In addition, the *RMSE* shows a gradually decreasing trend from the southeast to the northwest, which is similar to the spatial pattern of precipitation. This phenomenon is reasonable because the *RMSE* decreased with declining precipitation intensity.

Figure 7 shows the spatial distribution of *RB* between 3B43V7 and the observations collected by the rain gauge. *RB* shows positive values in the southeastern part of mainland China and negative values in the northwestern part. 3B43V7 heavily underestimates the precipitation over the humid coastal regions in Southeastern China but seriously overestimates the precipitation in the non-humid regions of the northwest and southwest. The results obtained here are consistent with *Liu et al. (2016)*, *Wu, Xu & Wang (2018)*, and *Yong et al. (2016)*. *Liu et al. (2016)* indicated that the underestimation was caused by the complex topographic conditions in the coastal areas that induce topographic rain; it was difficult for TRMM data to capture this kind of local precipitation event. However, the overestimation in northwest China is due to the complexity of its topography and

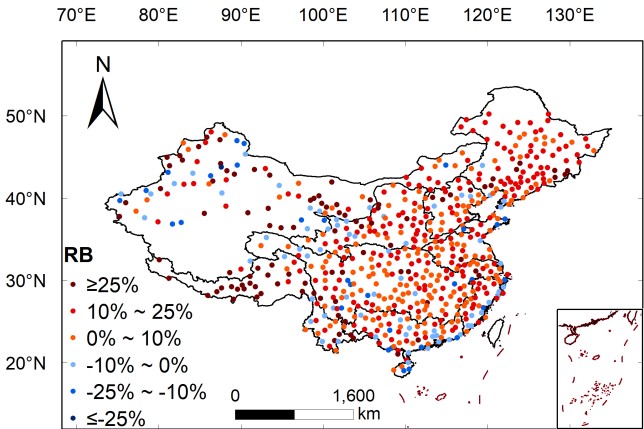

**Figure 7** Spatial distribution of *RB* between 3B43V7 product and rain gauge observations.

geomorphology and the small average precipitation, which has a great influence on the recorded error (*Tan & Duan, 2017*). *Yong et al. (2016)* identified a different conclusion, stating that the underestimation was mainly caused by missed precipitation, while the overestimation in non-humid regions came from hit bias and false precipitation. The specific reasons need to be further studied. In addition, *Villarini et al. (2008)* found that there was a tendency for the error to decrease for increasing number of rain gauges.

## Evaluation of errors on basin scale

Figure 8 shows the box plots of the statistic metrics over nine basins. There are large *RB* and *R* in the Inland River Basin and the Southwest River Basin. The values of the first and third quartiles and the upper and lower ends of the outliers are greater in the Inland River Basin and Southwest River Basin than those in other basins. A similar finding was reported by *Huang et al. (2016)*, who considered that the large biases in TMPA precipitation over western China may be attributed to the sparse ground gauge-based information adopted in the GPCC data for the bias adjustments. From the Southeast River Basin to the Inland River Basin, the *RMSE*'s values of first and third quartiles and upper and lower end of the outliers are gradually closer to the median. 3B43V7 and rain gauge observations show the most reliable *RMSE* in the Inland River Basin.

Figure 9 shows the average monthly precipitation and *RB* histogram of 3B43V7 and rain gauge observations from 1998 to 2016. The precipitation over the basin is concentrated from June to September and peaks in July, but is at a minimum in winter. 3B43V7 overestimates the rain gauge observations, and the *RB* is less than 16%. On the whole, 3B43V7 shows an acceptable detection capability, but with obvious differences among sub-regions. The *RB* of 3B43V7 in the Pearl River Basin and Southeast River Basin is the least, and the recordings made by 3B43V7 are even lower than the rain gauge observations during the dry season. It is clear that 3B43V7 overestimates the monthly precipitation estimates over the Southwest River Basin. 3B43V7 performs better in the wet periods than that in the dry periods in the Yangtze River Basin; these results are corroborated by *Jin, Zhang & Huang (2015)*. The

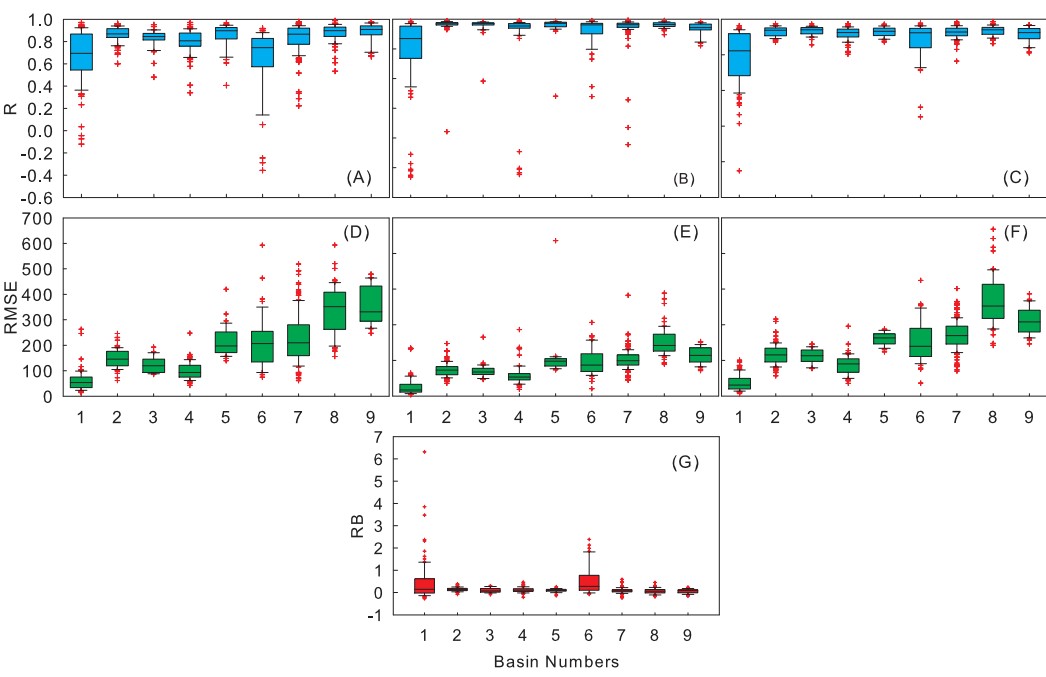

**Figure 8** Boxplots of *R* (A–C), *RMSE* (D–F) at the annual, seasonal, monthly scales and boxplots of monthly *RB* (G) between the 3B43V7 product and rain gauge observations.

*RB* presents an inverted arch shape with the bottom in summer. In addition, the *RB* is relatively high in the Yellow River Basin, Haihe River Basin, and the Songliao River Basin. Overall, 3B43V7 has an accurate presentation in the wet season with more errors in the dry season. A similar finding was reported by *Huang et al. (2016)*, who found that satellite estimates generally were better able to capture the overall spatial–temporal variations of precipitation over China in warm versus cold seasons. Meanwhile, 3B43V7 also tends to show better agreement with rain gauge observations over humid regions than over arid and alpine regions.

Figure 10 shows the annual variations of gauge observations and precipitation estimates from 3B43V7 during 1998 to 2016 in nine basins. Although 3B43V7 can capture the annual trend of precipitation the estimated precipitation value of 3B43V7 in each basin is greater than that observed on the ground. The annual amount of precipitation shows a decreasing trend in the Southwest River Basin and the Huaihe River Basin. This is consistent with the results made by *Zhu et al. (2017)* in the southwest rivers and is most obviously seen with the declining trend of precipitation in the Southwestern River Basin. The highest upward trend of precipitation is observed in the Pearl River and Southeast River Basin, and the variation in precipitation in the Yangtze River Basin is rising slowly. The annual precipitation observation of 3B43V7 is obviously different from those of the meteorological stations in the Inland River Basin, which is consistent with *Liu et al. (2016)*. Overall, 3B43V7 tends to overestimate annual precipitation over mainland China.
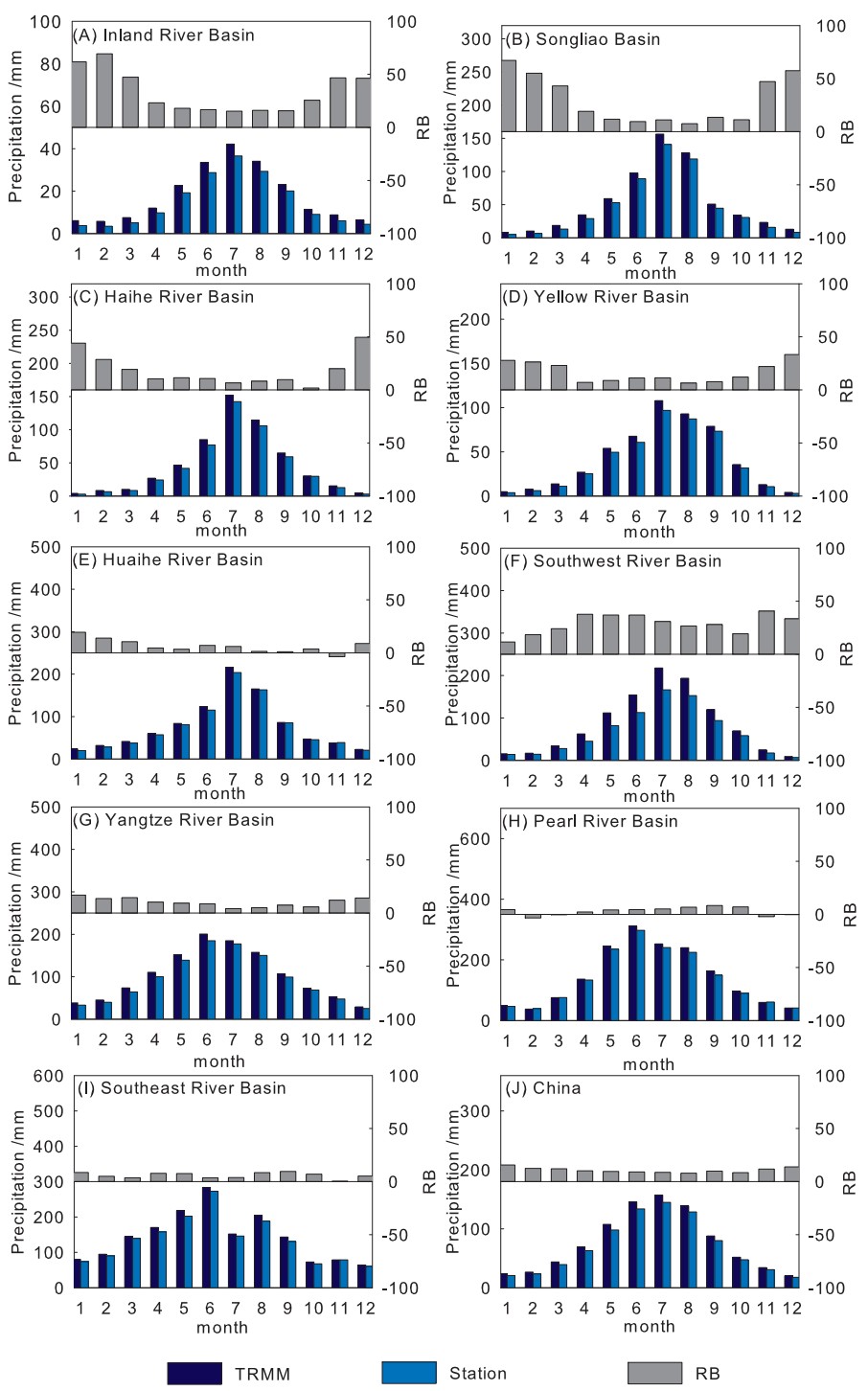

**Figure 9** Comparisons of mean monthly precipitation from the 3B43V7 product and rain gauge observations for each basin (A–I) and mainland China (J) during 1998–2016.

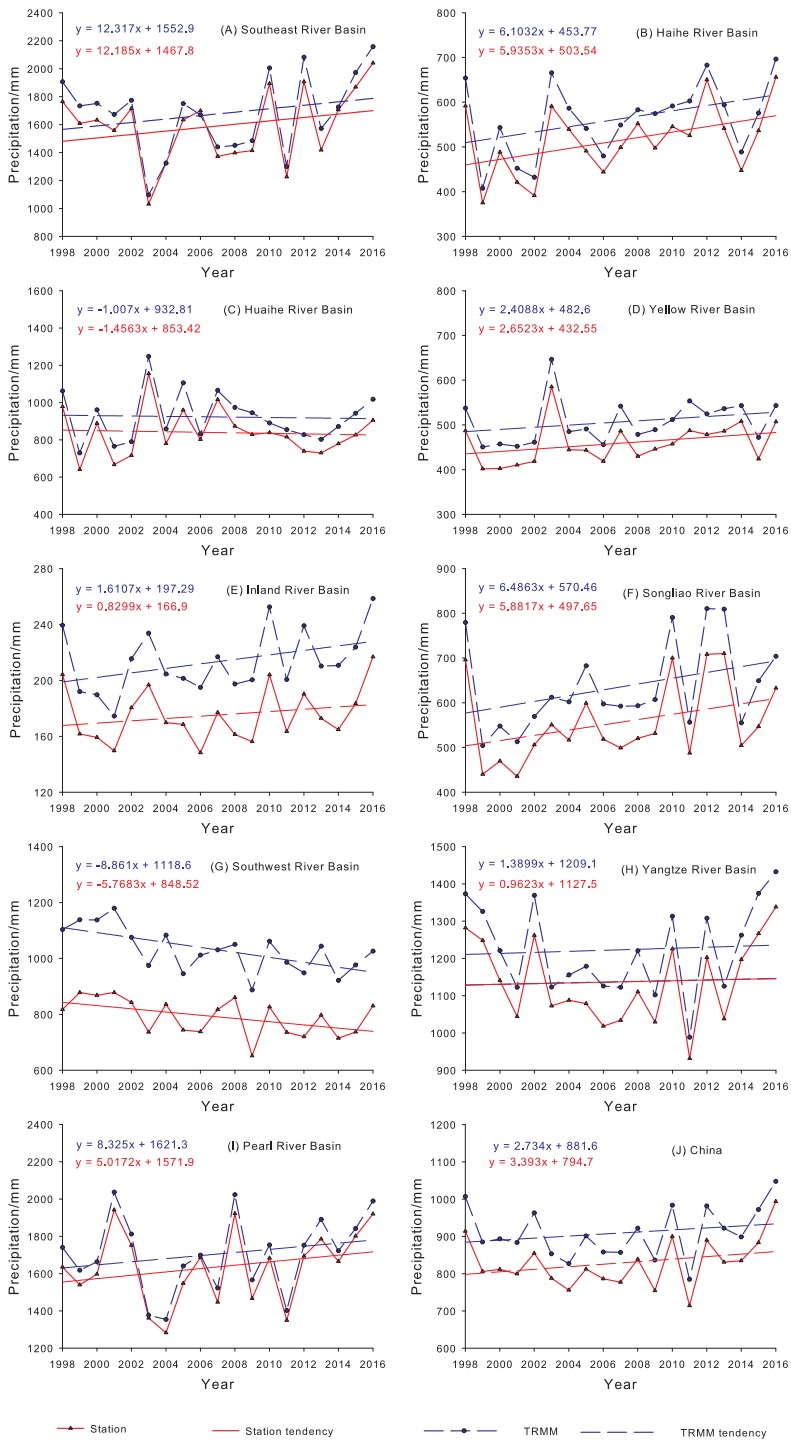

**Figure 10** **Variations of annual precipitation of the 3B43V7 product and rain gauge observations for each basin (A–I) and mainland China (J) during 1998–2016.**

## Evaluation of elevation-dependent error

The distinct topography of mainland China may cause terrain-induced errors on remote sensing retrievals and therefore, determining the effects of topography on the quality of data from 3B43V7 is indispensable. To assess the effects of topography on the performance of 3B43V7, all of mainland China was divided into ten elevation ranges: (A) <500 m, (B) 500–1,000 m, (C) 1,000–1,500 m, (D) 1,500–2,000 m, (E) 2,000–2,500 m, (F) 2,500–3,000 m, (G) 3,000–3,500 m, (H) 3,500–4,000 m, (I) 4,000–4,500 m and (J) 4,500–5,000 m. The scatterplots of the monthly precipitation estimates of 3B43V7 versus the rain gauge data at different elevation ranges are presented in Fig. 11. TRMM 3B43V7 presents a strong correlation with the gauge data when the elevation is less than 3,500 m. However, the $R$ between 3B43V7 and rain gauge data decrease at elevations of 3,500–4,500 m ($R$ less than 0.80), which may be caused by the climatological calibrations of 3B43V7 that eliminate the topographic effect (*Bolvin & Huffman, 2015*). Climate calibration is based on the data gathered from hundreds of basic meteorological stations in China, all of which are below 3,000 m above sea level, making 3B43V7 insensitive to terrain enhancement when the elevation is less than 3,000 m (*Bolvin & Huffman, 2015*). TRMM 3B43V7 has the lowest *NSE* at elevations of 3,500–4,500 m, indicating that it has large mean square differences from station observations. *RMSE* shows an opposite trend to *RB*. The highest *RMSE* appears in the 0–500 m elevation range (39.86 mm). *RB* increases with elevation (from 7.9% in 0–500 m to 57.3% in 4,000–4,500 m). It is consistent with the findings of *Jin, Zhang & Huang (2015)*, *Mantas et al. (2015)* and *Zhu et al. (2017)*, who found that the deviation between TRMM 3B43 and meteorological observation data showed a positive trend with the increase of elevation in the Yangtze River Basin and Southwest River Basin. *Yong et al. (2016)* also indicated that the bias in TMPA gradually increased toward the higher elevations of mainland China. In addition, *Guo et al. (2016)* found that significant elevation-dependent errors existed in TMPA over central Asia, especially for the high-elevation regions, which is consistent with our results.

## CONCLUSIONS

The accuracy of 3B43V7 over mainland China was examined using rain gauge observations as a reference. The evaluation was conducted by analyzing the spatial and temporal distribution of precipitation based on *R*, *RMSE*, *RB* and *NSE*. We also focused on examining the performance of 3B43V7 and its dependence on topographical factors. The major conclusions are presented as follows:

(1) The precipitation estimated by 3B43V7 is well-correlated with station data on all timescales (annual, seasonal and monthly). The *R* and *NSE* between 3B43V7 and rain gauge observations increases with temporal aggregation. Monthly estimates from 3B43V7 have a good agreement with gauge observations in terms of precipitation magnitude. On a seasonal scale, the poorest *RB* (14%) for 3B43V7 was observed in the winter but it has the lowest *RMSE* (7.60 mm). On the contrary, the lowest *NSE* (0.82), the lowest *RB* (8.7%) and highest *RMSE* (33.44 mm) were obtained in summer. In addition, 3B43V7 has a clear overestimate each month. More than 10% of *RB* occurs from November to April. 3B43V7

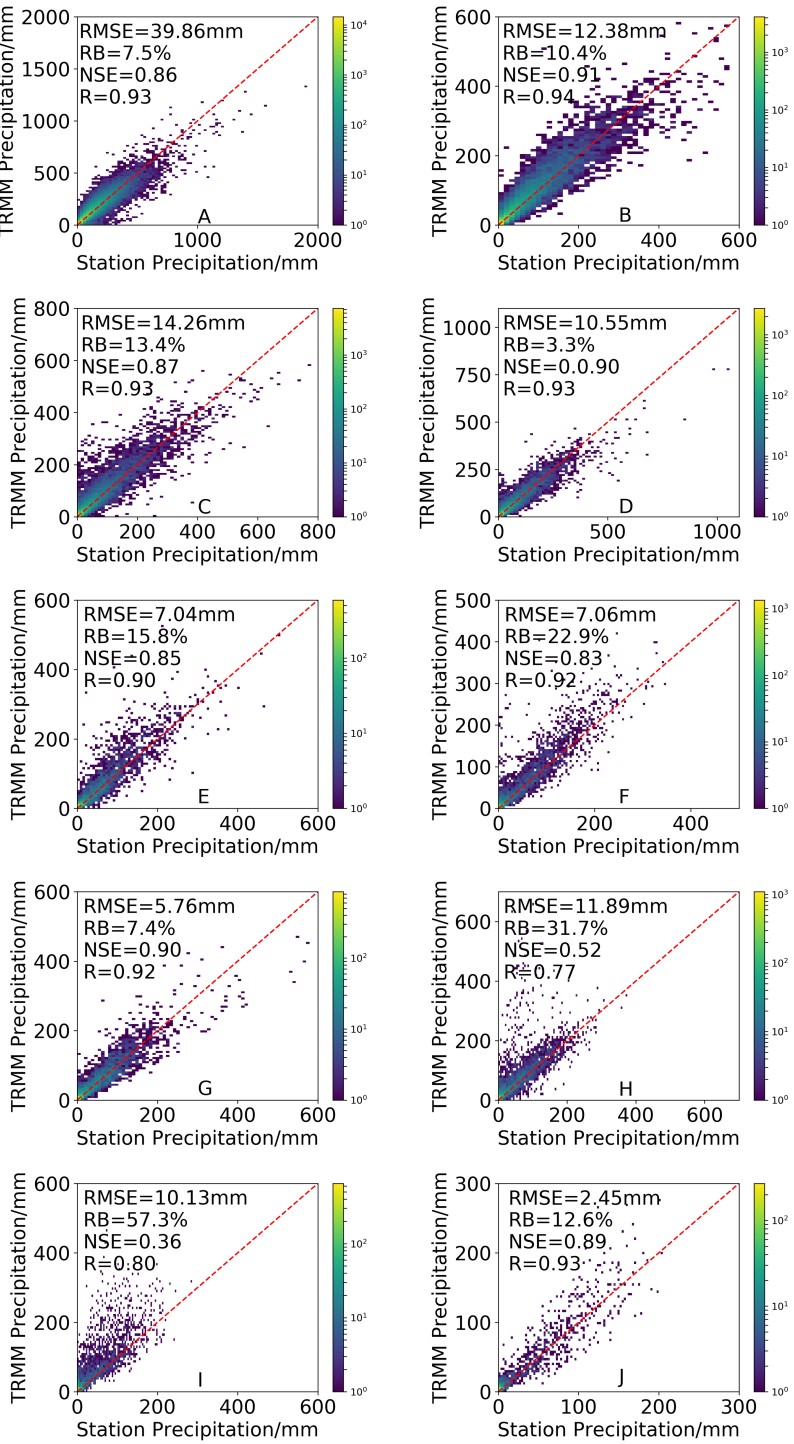

**Figure 11** Density-colored scatterplots of the 3B43V7 product versus rain gauge observations at different elevation ranges. All results have passed the significance test.

presents well in wet seasons with more errors in the dry seasons. However, there is a better consistency between 3B43V7 and the precipitation observed by conventional rain gauges during dry period**s**.

(2) The spatial trend of *R* is similar to *RMSE* and opposite to *RB*, with 3B43V7 performing better over southeastern China (such as the Southeast River Basin and Pearl River Basin) compared with northwestern China (such as the Inland River Basin and Southwest River Basin). The amount of precipitation strongly affects the performance of 3B43V7 over the study area with greater errors and lower correlations and *RMSE* in regions with very low precipitation. The *RB* is relatively high in the Yellow River Basin, Haihe River Basin, and Songliao River Basin. The annual precipitation showed a decreasing trend in the Southwest River Basin and Huaihe River Basin, although the declining trend of precipitation in the Southwestern River Basin was the most obvious. The highest upward trend of precipitation was observed in the Pearl River Basin and Southeast River Basin, and the variation of precipitation in the Yangtze River Basin increased slowly.

(3) The correlation decreased rapidly when the elevation was greater than 3500 m. *RB* estimates increased with elevation, and the worst *RB* and *NSE* value appeared at $4000 \leq 4500$ m. The *RMSE* decreased with elevation, and the highest *RMSE* appeared in the 0–500 m elevation range.

The results of this study will give important information about the promises and shortcomings of the product in the study area and in similar regions. An important research focus for future work will be combining the respective advantages of TRMM products and rain gauge observations to estimate more accurate near-real-time precipitation.

## ACKNOWLEDGEMENTS

The authors would like to express appreciation to the editors and reviewers for their help.

### Funding

This work was supported by the National Natural Science Foundation of China (41807170, 41601600), the Major Science and Technology Innovation Projects of Shandong Province (2019JZZY020103), the Natural Science Foundation of Shandong Province (ZR2017BD021), the SDUST Research Fund (2014TDJH101), the Opening Fund of Key Laboratory of Geomatics and Digital Technology of Shandong Province, Graduate Technology Innovation Project of SDUST (SDKDYC190305), the Applied Basic Research Project of Qingdao (18-2-2-42-jch), the Guizhou Provincial Education Department Innovation Group Major Research Project (KY [2016] 055, 054) and the Doctor Star-up Foundation of Qingdao Agricultural University (663/1117012). The funders had no role in study design, data collection and analysis, decision to publish, or preparation of the manuscript.

### Grant Disclosures

The following grant information was disclosed by the authors:

National Natural Science Foundation of China: 41807170, 41601600.
Major Science and Technology Innovation Projects of Shandong Province: 2019JZZY020103.
Natural Science Foundation of Shandong Province: ZR2017BD021.
SDUST Research Fund: 2014TDJH101.
Opening Fund of Key Laboratory of Geomatics and Digital Technology of Shandong Province.
SDUST: SDKDYC190305.
Applied Basic Research Project of Qingdao: 18-2-2-42-jch.
Guizhou Provincial Education Department Innovation Group Major Research Project: KY [2016] 055, 054.
Doctor Star-up Foundation of Qingdao Agricultural University: 663/1117012.

## Competing Interests

The authors declare there are no competing interests.

## Author Contributions

- Ziteng Zhou conceived and designed the experiments, performed the experiments, analyzed the data, prepared figures and/or tables, and approved the final draft.
- Bin Guo conceived and designed the experiments, authored or reviewed drafts of the paper, and approved the final draft.
- Youzhe Su performed the experiments, authored or reviewed drafts of the paper, and approved the final draft.
- Zhongsheng Chen conceived and designed the experiments, authored or reviewed drafts of the paper, and approved the final draft.
- Juan Wang performed the experiments, authored or reviewed drafts of the paper, and approved the final draft.

## Data Availability

The raw data are available in the Supplemental Files.

## Supplemental Information

Supplemental information for this article can be found online at http://dx.doi.org/10.7717/peerj.8615#supplemental-information.

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
