# Peer review of "Multidimensional evaluation of the TRMM 3B43V7 satellite-based precipitation product in mainland China from 1998–2016"

_PeerJ, doi:10.7717/peerj.8615_

## Round 0.1 · original submission · Major Revisions

The reviewers have made many helpful suggestions to guide your revision. Please go over their responses carefully, and respond to their comments. I agree with both reviewers that the use of only RMSE and R2 provides only a simplistic look at the statistical relationships. Using Nash-Sutcliffe (such as in Krakauer, N.Y.; Pradhanang, S.M.; Lakhankar, T.; Jha, A.K. Evaluating Satellite Products for Precipitation Estimation in Mountain Regions: A Case Study for Nepal. Remote Sens. 2013, 5, 4107-4123 or many other studies) would add value to the study.

Reviewer 1 ·

Basic reporting

The paper with NO. peerj-40927 focused on the evaluation of TRMM 3B43V7 with ground observations over mainland China. This work was conducted based on three traditional metrics (R values, RMSE and RB). The paper is generally well written and the results are useful for the improvement of algorithms and applications of 3B43V7. However, the study is regular without enough novelty. I suggest a major modification. Here are some comments may be useful for the improvement of this paper.

Experimental design

The evaluation metrics were widely used and traditional. It is some kind too simple to provide enough evaluation information by using just R, RMSE and RB. The evaluation work was conducted at multiple scales which could provide useful reference for the readers of this journal.

Validity of the findings

The results are validated. However, the independence between rain gauges from CMA and rain gauges used for generating TRMM 3B43V7 product. In addition, some redundant sentences in the findings should be reduced to make the text be brief.

Additional comments

1. In the study, the grids including at least one rain gauge from TRMM 3B43V7 were extracted and compared with the ground-based rain gauges from CMA. However, part of these observed gauges from CMA have been incorporated in the generation of TRMM 3B43V7 as mentioned: “Some of the data from certain stations utilized in this present study could also be used for bias adjustment of the TRMM 3B43V7 product”. Therefore, the independence between rain gauges from CMA and rain gauges used in generating 3B43V7 should be analyzed in the dataset section. This is very important for the accuracy of the evaluation results. Otherwise, the evaluation results can hardly be to convince.
2. The caption of figures and tables cannot be found in the text which make the review inconvinient.
3. Some result sentences are repeated. For example, the authors analyzed results by repeating the statistical values (e.g. R values, RMSE values). For saving space, please remove the repeated sentences which can easily be obtained from the figures. For example: “The R values at annual, seasonal, and monthly timescales were 0.96, 0.95, and 0.93, respectively”; “The values from RMSE at annual, seasonal, and monthly timescales were 54.56 mm, 41.79 mm, and 34.86 mm, respectively”; “It is obvious that 3B43V7 had higher R values of 0.96, 0.92, 0.92, and 0.95 for spring, summer, autumn, and winter, respectively.”….
4. Datasets: I suggest the authors improve the dataset section with great modifications by considering the following aspects:
(1) Give a clear and detailed description for 3B43V7 datasets especially the generation processes and the correction process by using rain gauges.
(2) More information on the benchmarked dataset (rain gauges from CMA) should be clearly described such as the background, the generation processes, the quality check processes, and the potential errors.
(3) Independence analysis between rain gauge from CMA and rain gauges used in generating 3B43V7 should be added.
5. The words should be well re-considered, like precipitation and rainfall. For example, in the text of the study area, the authors used “annual precipitation”, however, used “rainfall” in the legend of Fig. 1.
6. English should be improved by native speaking people in this area.
7. Lines 47-59: The sentence is not clear. Reword it.
8. English problems: The English should be improved greatly with polishing by local speaking experts.
9. Lines 60-61: what’s your evidence to conclude that “the TMPA precipitation estimate is considered to be one of the most reliable and widely-utilized products”?
10. Line 63: why you evaluate the product again even though the TRMM had retired since several years ago?
11. Lines 73-99: The authors left a large space for the previous studies on evaluation of TRMM products by listing the main findings. However, it is not clear and easy comparable enough for the readers. The following aspects should be improved for this part: (1) the historical studies should focus on the 3B43 product studied in this study rather than all products from TMPA algorithm (e.g. 3B42V7, 3B42V5, 3B42RTV7) as the performance for different versions of TRMM TMPA algorithm vary greatly due to its different inherent algorithms. In other words, the performances between different versions cannot be compared with the same standard. (2) Actually, the product TRMM 3B43V7 has been evaluated by using different methods over different regions (including mainland China, for example (Chen and Li, 2016)) compared with other satellite-based precipitation estimates or even improved by using different methods. The authors should pay more attention to these studies. To provide a better background knowledge, I suggest the authors compare these previous studies in one table with time, area, standard data, period, main findings, references. (3) after adding one previous study table, the novelty or at least the difference between the current study and the listed previous studies should be highlighted.
12. The significance of this study is too general or not specific by just reading as “The results of the study can provide a basis for hydrological research and water resource management”. In addition, daily datasets may be more frequently used in hydrological research. Therefore, the significance of this evaluation work should be re-considered and highlighted with more specific applications like drought monitoring.
13. Study area: Why did the authors divide the mainland China into 9 basins? The original purpose should be mentioned in the study area section. In addition, if the authors intend to use these 9 basins as evaluation sub-regions, the introduction of these basins should be added. For saving space, a table could be used here.
14. Lines 132-133: I cannot agree with “This satellite mission obtained the longest data series of almost the total amount of global precipitation”. The data records of TRMM is not the longest one.
15. The sentences are not accurate, the latest version of the TRMM TMPA algorithm is Version 7, while the version 7 products include different estimates like 3B42V7, 3B42RTV7, 3B43V7.
16. Line 160: Based on the “relatively complete measurements were selected”, I can guess that part of the 620 gauges selected is not complete. The accurate standard for selecting gauges from 756 gauges should be given here. In addition, the quality check should also be mentioned like what is the type of rain gauges for measuring rainfall. Could the snowfall be also measured by the rain gauges?
17. Line 194: remove the dot.
18. Line 201: replace “higher” with “high”.
19. Line 205: replace “(with RB of 14.0%)” by “(RB=14.0%)”
20. Line 207: “This may be the result of snowfall, which is a factor that seriously affects the precipitation estimates made by satellites in North China (Chen & Li 2016).” Could the rain gauges from CMA capture snowfall? What kind of rain gauges used for capturing snowfall? The answer to these questions should be mentioned in the dataset section.
21. Line 297: modify the title as “Evaluation of elevation-dependent error”
22. Line 301-303: why only four classes used in the analysis? Is there any standard to split the elevation range into “<500 m, 500-1500 m, 1500-3000 m, and >3000 m”? I suggest the authors split elevation range into more number classes, which will be more useful for application users of 3B42V7.

References
Chen, F.R., Li, X., 2016. Evaluation of IMERG and TRMM 3B43 Monthly Precipitation Products over Mainland China. Remote Sensing, 8(6): 472. DOI:10.3390/rs8060472

Reviewer 2 ·

Basic reporting

The manuscript needs some improvement in English. Generally, the manuscript organization is good. The figures and table are well made.

Experimental design

I suggest the authors to add a strong statistical metric, for example, Nash-Sutcliife Efficiency. The metrics the authors used in the manuscript are weak in terms measuring the performance of the TRMM data as well as pointing out the causes of the bias in the data.

Validity of the findings

The statistical tests needs to be supplemented by significance test results.

Additional comments

I have provided detail comments/changes and suggestions annotated with the pdf of the manuscript attached.

Annotated reviews are not available for download in order to protect the identity of reviewers who chose to remain anonymous.

---

## Round 0.2 · Minor Revisions

The revised version of your manuscript has been substantially improved. However, there is one point that reviewer 1 has made that though it is moderately addressed in the rebuttal letter is not addressed in the manuscript itself, regarding the Independence of measurements. As suggested by the reviewer, this should either be explained in the manuscript, or a "worst case" comparison be made by excluding all grid cells with known rain gauges.

In addition, it is unclear what the modified sentence in L49-50 intends to say, and this should be re-written.

Reviewer 1 ·

Basic reporting

The paper has been well improved.

Experimental design

The independence problem should be explained further.

Validity of the findings

None

Additional comments

The paper with NO. peerj-40927-v1 focused on the evaluation of TRMM 3B43V7 with ground observations over mainland China. The paper is improved after the first round modification. Most comments have been responded well except one. Therefore, I suggest a minor modification. Here are some comments may be useful for the improvement of this paper. The response for independence problem is not acceptable by listing some references or “discussed with several famous experts in this fields”.
The independence problem should be mentioned in the manuscript. I can understand that “the station name and station number cannot be obtained”, but you can obtain the grids contains “the number of sites in each box”. As we all know that the TRMM 3B43V7 is corrected based on GPCC at a monthly scale. In China mainland area, there is a possibility that the dependence could be high to make the validation results unstable. According to CMA, at least 160 international stations are distributed over mainland China. In my opinion, the authors could assume that all repeated grids containing gauges in GPCC are repeated with gauges you used as reference. In this case, if the 3B43V7 still have a consistent performance, then the authors could conclude that “the results are credibility”. Otherwise, the authors should explain the independence problem in the manuscript for the readers.

Reviewer 2 ·

Basic reporting

The writing is much improved. Figures and Tables are organized and clear.

Experimental design

No further comments.

Validity of the findings

No further comments.

---

## Round 0.3 · accepted · Accept

Thank you for making the modifications to update the manuscript to address the independence issue related to the unknown overlap of rain gauges between data sets.